# Wide-Area Visual Monitoring System Based on NB-IoT

**DOI:** 10.3390/s25216589

**Published:** 2025-10-26

**Authors:** Guohua Qiu, Weiyu Tao, Rey-Chue Hwang, Chaofan Xie

**Affiliations:** 1School of Electronic and Mechanical Engineering, Fujian Polytechnic Normal University, Fuzhou 350300, China; qghly@fpnu.edu.cn (G.Q.);; 2Modern Facility Agriculture Engineering Research Center of Fujian Colleges and Universities, Fujian Polytechnic Normal University, Fuzhou 350300, China; 3Department of Electrical Engineering, I-Shou University, Kaohsiung 84001, Taiwan; 4School of Big Data and Artificial Intelligence, Fujian Polytechnic Normal University, Fuzhou 350300, China

**Keywords:** detection, monitoring, NB-IoT, CoAP Protocol, MQTT Protocol, image

## Abstract

**Highlights:**

An integrated architecture for real-time event detection and response has been proposed, addressing the critical challenge of effective detection of unexpected events in wide-area surveillance. Utilizing the wide coverage of narrowband cellular networks (e.g., NB-IoT), the system ensures efficient image data transmission even in remote or rural areas, providing real-time alerts.

**What are the main findings?**
The system detects abnormal events by analyzing sequential image frames using intelligent algorithms and stores images only upon anomaly detection, improving storage efficiency.By using the CoAP protocol to transmit encapsulated JPEG images and leveraging the MQTT protocol to deliver image data to client applications, the system achieves efficient data transmission and processing.

**What is the implication of the main finding?**
This study offers an intelligent, scalable, and responsive solution for wide-area surveillance systems, overcoming limitations of traditional systems such as low storage efficiency, limited transmission range, and complex operation.The intelligent anomaly detection algorithms reduce the risks and costs associated with manual monitoring, enhancing both the efficiency and accuracy of anomaly detection.

**Abstract:**

Effective detection of unexpected events in wide-area surveillance remains a critical challenge in the development of intelligent monitoring systems. Recent advancements in Narrowband Internet of Things (NB-IoT) and 5G technologies provide a robust foundation to address this issue. This study presents an integrated architecture for real-time event detection and response. The system utilizes the Constrained Application Protocol (CoAP) to transmit encapsulated JPEG images from NB-IoT modules to an Internet of Things (IoT) server. Upon receipt, images are decoded, processed, and archived in a centralized database. Subsequently, image data are transmitted to client applications via WebSocket, leveraging the Message Queuing Telemetry Transport (MQTT) protocol. By performing temporal image comparison, the system identifies abnormal events within the monitored area. Once an anomaly is detected, a visual alert is generated and presented through an interactive interface. The test results show that the image recognition accuracy is consistently above 98%. This approach enables intelligent, scalable, and responsive wide-area surveillance reliably, overcoming the constraints of conventional isolated and passive monitoring systems.

## 1. Introduction

Initially, electronic surveillance systems operated using analog signal transmission, commonly referred to as Closed-Circuit Television (CCTV). These systems predominantly employed analog matrix control and were constrained by several limitations, including short transmission distances, high operational complexity, limited storage capacity, and susceptibility to electromagnetic interference [1]. In the 1990s, the introduction of Digital Video Recorder (DVR) technology enabled the digitization and storage of analog video signals on high-capacity hard drives, significantly improving system capabilities. DVR systems offered enhanced video data retention, extended transmission ranges, and improved operational convenience compared to traditional analog solutions [2].

By the late 20th century, the proliferation of IP Video Surveillance (IPVS) systems, which leveraged internet-based technologies, markedly expanded surveillance coverage beyond earlier generations. These systems enabled integration with various platforms, utilized disk array architectures to increase storage capacity, and incorporated fault tolerance mechanisms through redundant network device configurations and failover protocols [3,4]. In recent years, the application of image processing techniques combined with intelligent algorithms has driven the evolution of intelligent surveillance technologies, resulting in increasingly large-scale, intelligent, and networked video monitoring systems [5,6,7]. As surveillance systems are deployed across a broader range of industries, public reliance on such technologies to ensure industrial and residential security continues to rise, accompanied by growing expectations for intelligent functionality.

With the emergence of the 5G era, Narrowband Internet of Things (NB-IoT) technology has laid a foundational framework for massive Machine-Type Communications (mMTC) within 5G networks. Its core advantage lies in providing efficient, reliable connectivity for a large number of low-power, low-cost, and widely distributed IoT devices. The benefits of NB-IoT modules are well-established [8], including near-complete nationwide base station coverage, narrow bandwidth utilization (180 kHz), and exceptionally low power consumption—averaging just 0.3 μA in Power Saving Mode (PSM) under a reference voltage of 3.3–5 V. Although Low-Power Wide-Area (LPWA) services have existed since the 2G era, global expansion revealed several challenges, including excessive terminal power consumption, high data transmission volumes, limited coverage, and elevated operational costs. Next-generation NB-IoT technologies effectively address these limitations.

Moreover, video surveillance systems inherently involve processing large volumes of image data. The integration of intelligent algorithms has begun to resolve long-standing issues associated with manual monitoring—such as “footage exists but anomalies are hard to identify,” “anomalies can be found, but detection is time-consuming,” and “services are available, but the cost is prohibitive.” These innovations also mitigate the risk of missed anomalies due to human oversight.

Therefore, in this paper, we propose an intermittent image capture approach, wherein intelligent algorithms detect anomalies through comparative analysis of sequential image frames. Images are stored only upon detection of an anomaly. Utilizing the extensive coverage provided by narrowband cellular networks (e.g., NB-IoT), the proposed system enables efficient image data transmission even from remote or rural areas, providing users with real-time alerts. Our experimental results show the transmission success rate achieves 100% at a baud rate of 115,200 bps. When faced with high-concurrency request volumes, the server maintains a 100% success rate and the image recognition accuracy is consistently above 98%. This paper provides a practical example and effect verification for the video detection application of NB-IOT, a new generation of low-power wide-area network communication technology.

The remainder of this paper is organized as follows. The design and construction of the system’s three-tier architecture will be analyzed in Section 1, with a focus on the proposed solution. Experimental testing is detailed in Section 2, and our conclusions are presented in Section 3.

## 2. Related Work

Based on the new generation of low-power wide-area network (WAN) IoT technology, remote detection of various parameters is extensively researched. For instance, the literature [9] presents the design of a smart city environment monitoring and optimization system based on NB-IoT technology. This system utilizes NB-IoT communication to acquire environmental monitoring data from the wide-area environment. Finally, the monitoring data are input into a BP neural network enhanced by the particle swarm optimization (PSO) method for environmental risk prediction. Similar studies can also be found in the literature [10,11,12,13,14]. Additionally, some scholars have conducted research on detecting data and performing control operations based on the NB-IoT system. The literature [15] introduces an intelligent street lamp control system based on narrowband internet of things (NB-IoT) technology. This system can automatically or remotely adjust the brightness and switch of street lamps according to demand and environmental conditions, while also monitoring and recording the current, voltage, power, and other data of street lamps. Similarly, the NB-IoT-based monitoring system for UAV networks presented in the literature [16] also tackles the challenge posed by the absence of a global IP address in the existing NB-IoT infrastructure. In recent years, research has progressively expanded into the realm of video detection. A video surveillance unit (VSU), as introduced in the literature [17], incorporates a motion detection function. Upon detecting motion within the camera’s field of view, images are captured, processed, compressed, and segmented to ensure they fit within the maximum payload size of LoRaWAN for transmission. In the literature [18] on smart IoT-based mobile sensors, a unit is used to collect information about the cane user and the surrounding obstacles while on the move, and an embedded machine learning algorithm is developed to identify the detected obstacles and alarm the user about their nature. There is also literature [19] focusing on video-based passenger counting systems. However, research in this area, similar to other studies [18,19], has not yet been applied to wide-area scenarios. In addition, some scholars specialize in the processing of video images. The literature [20] presents enhancement techniques and synthetic image generation methods utilizing YOLO, SSD, and EfficientDet deep learning models to enhance sea mine detection technology. The literature [21] introduces a searchable and revocable attribute-based encryption scheme specifically tailored for dynamic video anomaly detection scenarios, enhancing the security and privacy of video data. The literature [22] explores and develops image compression and data transmission methods that contribute to achieving stable low-rate transmission of images and data in Internet of Things (IoT) systems. Some of these studies are limited to the detection and management of environmental parameters, failing to fully reflect the on-site visual scene. Others are focused on the research of deep learning algorithms for video images, which is not compatible with remote narrowband IoT communication technology. Therefore, the proposed system combines IoT technology and optimized background subtraction method to detect moving targets, making it the most reasonable choice for video surveillance in a wide-area environment.

## 3. System Design

### 3.1. Subsection

The proposed system adopts a three-tier architecture comprising: (i) the embedded and sensor perception layer, (ii) the network layer utilizing narrowband cellular communication, and (iii) the Web backend service and application layer, integrating functionalities from the backend to the frontend [23,24,25].

First, the sensing layer (Embedded Devices & Sensors) is built around an ARM Cortex-M4 master chip, which acquires image data from an camera sensor via the RS232 protocol.

Next, the network layer (Narrowband Cellular Communication) employs NB-IoT for connectivity. In this stage, the NB module receives hex-encoded JPEG image data processed by the ARM processor and transmits it transparently to the server using the CoAP protocol, with data forwarding handled by base stations [26].

Finally, the service/application layer (Web Backend) is implemented using a Spring Boot-based microservices framework with a Vue.js frontend. The system integrates Eureka for service registration, Feign for service invocation, and Nginx for load balancing and IoT data collection. Image data is delivered via WebSocket, decoded, stored in a distributed MySQL cluster managed by Mycat, and made accessible through RESTful and Webservice APIs to support real-time monitoring and anomaly detection.

The design of the system software encompasses the development of encoding algorithms for raw image data, serial communication protocols, and communication interfaces among NB-IoT modules, base stations, and IoT servers, as well as the interaction between the Web service backend and IoT servers. Additionally, it includes the design of image encoding and decoding algorithms, as illustrated in Figure 1. The system initially captures image data at predefined, fixed intervals and encodes these images into hexadecimal format. This data is then transmitted via a serial port to the STM32F407 main controller, which subsequently forwards it to the NB-IoT module through the same serial interface. The NB-IoT module relays the data packets to the IoT service backend via the NB communication base station. Upon receipt, the backend processes the data using the business logic implemented in the Service layer and employs the Spring Data JPA persistence framework to store the processed results in a MySQL database. Finally, the frontend—developed with Vue.js—sends requests to the Spring Boot backend to retrieve the stored data, which is then rendered visually on the user interface. In the case of abnormal image detections, the images are archived, and users are promptly notified.

### 3.2. System Perception Layer

The system hardware primarily comprises a wide-area visual monitoring setup, consisting of four core components: an OV7725 camera sensor(Omnivision, Shanghai, China), an STM32F407ZET6 processor(STMicroelectronics, Lausanne, Switzerland), an NB-IoT communication module(China Mobile IOT, Chongqing, China), and a solar-powered energy supply unit. The OV7725 camera sensor captures image data at scheduled intervals and transmits it to the ARM-based processor for subsequent analysis. The perception layer relies on solar energy for power supply. During the day, the solar energy system charges the solid-state energy storage capacitor while also supplying power to modules such as the ARM processor, camera, and NB-IOT. At night, the solid-state energy storage capacitor releases stored energy to provide the electrical power required for the perception layer to operate.

#### 3.2.1. Hardware Architecture of the System Perception Layer

In the proposed design, the STM32F407ZET6 microcontroller, based on the ARM Cortex-M4 core, is selected as the central processing unit. Communication with the NB module is established via an RS-232 serial interface, while the OV7725 camera sensor interfaces with the main control unit through a USART serial connection. Furthermore, the system is powered by a solar energy supply coupled with an energy storage module. The hardware architecture of the system is illustrated in Figure 2.

#### 3.2.2. Design of the Hardware Circuitry for the Perception Layer

NB-IoT Module

The data on the maximum power consumption and transmission ranges of various communication modules are presented in Table 1. In wireless wide-area communication, achieving longer transmission distances inherently requires higher power consumption due to propagation losses. These communication technologies also vary in terms of latency and cost, and the best choice depends on the specific application scenario requirements. In the current IoT industry, there is a pressing demand for both extended communication ranges and cost-effective terminal solutions. These requirements can only be met by next-generation IoT chipsets, such as NB-IoT and LoRa, which are specifically designed to balance long-distance connectivity with low power consumption. Both NB-IoT and LoRa belong to the new generation of low-power wide-area network communication technologies, and their maintenance current in low-power working mode is only about 1 microampere. In terms of transmission distance, NB-IoT compensates for its coverage shortcomings through base station density, achieving coverage up to 35 km in suburban areas, but it relies on the coverage range of 4G/LTE networks. LoRa achieves a maximum single-point coverage of up to 15 km, but this is achieved through spread spectrum technology, which directly results in the maximum transmission rate typically below 50 kbps, much lower than the 250 kbps of NB-IoT communication. A comprehensive comparison reveals that LoRa is more suitable for low-data-volume, long-distance application scenarios such as agricultural monitoring and smart manhole covers, while NB-IoT is more suitable for frequent data interaction or low-latency requirements such as smart meter readings and real-time monitoring.

NB-IoT (Narrowband Internet of Things) is a dedicated communication network developed on the basis of cellular wireless infrastructure. It was introduced and promoted by network operators and telecommunications equipment manufacturers [27]. Operating within the licensed frequency bands of cellular networks, NB-IoT ensures stable and secure communications, effectively minimizing interference from other wireless devices. It supports CoAP protocol-based data transmission over extensive geographic areas [28,29,30]. When integrated with existing LTE network spectrum resources, NB-IoT can significantly reduce hardware costs. As an emerging wide-area network and wireless communication access solution, it is well-suited for large-scale deployment across diverse regions. From the very beginning of its standard design, NB-IoT technology has fully considered large-scale, low-cost, and low-power IoT deployment scenarios. In terms of handling network congestion, we adopt PSM/eDRX signaling reduction technology for prevention, and optimized RACH, multi-carrier, ACB/EAB access control, and QoS priority scheduling technologies for mitigation. In terms of handling latency, we utilize control plane optimization methods for fast transmission of small data packets, a power-saving mode that sacrifices real-time performance for delay tolerance, and priority scheduling technology to ensure low latency for critical services. In terms of ensuring QoS, we adopt differentiated services based on QCI, intelligent wireless resource scheduling, dedicated bearers in the core network, and retransmission to ensure high-reliability transmission. For example, In Power-Saving Mode (PSM), the system enters a deep sleep state, maintaining only essential active components, such as the clock, with microampere-level current consumption, thereby minimizing energy usage. During this state, the NB-IoT module remains registered on the remote CoAP network.

The peripheral hardware of the NB module in the system design includes a power supply unit, a status indicator LED, a reload button, a reset button, an antenna, and a SIM card. The baud rate for serial communication between the NB module and the main control MCU is configured to 9600. The schematic diagram of the NB module’s peripheral circuitry is presented in Figure 3. It is important to note that, when designing the antenna interface, a π-type impedance matching network is required for proper antenna connection. Additionally, the power supply voltage should be maintained as close as possible to the recommended 3.7 V to ensure optimal performance of the NB module.

STM32F407ZET6 Processor

The STM32F4 series, developed by STMicroelectronics, is a high-performance microcontroller family based on the ARM Cortex-M4 core, featuring a CPU clock frequency of up to 168 MHz. This series incorporates single-cycle DSP instructions and an integrated floating-point unit (FPU), significantly enhancing computational performance. The design requires high-speed serial communication, and the USART interface of the STM32F4 series supports data transmission rates of up to 10.5 Mbit/s. Compared with the STM32F1 series, the STM32F4’s serial interface enables reliable communication and data exchange with a camera module’s serial port, achieving an accuracy rate of up to 100%. This effectively resolves the issue of incomplete image data caused by data loss due to the lower serial transmission rates in the STM32F1 series [31,32]. Early testing revealed that when the USART3 of the STM32F103ZET6 rapidly received hexadecimal data transmitted by the OV7725 image sensor, the error and loss rate reached as high as 80%.

The peripheral circuit of the main control unit STM32F407ZET6 is shown in Figure 4. The two serial ports, USART1 and USART3, communicate with the NB module and the OV7725 camera respectively via serial communication, with baud rates set to 9600 and 115200. Upon triggering a single capture command via the button, the processor transparently transmits the data to an IoT terminal supporting the CoAP protocol, with an LED indicating the transmission status.

OV7725 Camera Sensor

Currently, CMOS cameras are extensively employed across diverse application scenarios, including products from manufacturers such as Hynix, Micron, and Samsung. The OV7725 camera module, developed by SmartView, supports continuous video output at resolutions up to 640 × 480 pixels at 60 Hz, offers high sensitivity, and incorporates a standard SCCB interface. It is equipped with multiple advanced features, including automatic edge enhancement, adaptive noise adjustment and suppression, frame synchronization modes, automatic band-pass filtering, and automatic white balance, all of which can be configured as required [33,34]. In this design, the OV7725 camera is selected for surveillance image acquisition, utilizing the RS232 communication standard to interface with the STM32F407 main control core unit. A single-shot command is executed to capture instantaneous images, which are then obtained in JPEG format.

#### 3.2.3. Design of Image Encoding

The system employs the OV7725 camera sensor to capture raw data in JPEG format. The communication protocol specifications for the OV7725 camera sensor are summarized in Table 2, with the communication baud rate configured at 115,200 bps.

The system processor communicates with the OV7725 camera sensor via serial communication in accordance with the specified protocol, transmitting corresponding data in hexadecimal format and processing the received data. The hexadecimal representation of JPEG image data begins with 0xFF and 0xD8 and terminates with 0xFF and 0xD9.

The communication process between the processor and the OV7725 camera sensor is illustrated in Figure 5, and the corresponding communication flowchart is presented in Figure 6. Initially, a hexadecimal command for a single-frame capture instruction is transmitted via the SCCB interface (see Table 2 for details). Upon receiving this command, the OV7725 immediately transfers image format data to the STM32F407 processor. Once the processor detects that the camera has completed the single-frame capture, it issues the command to retrieve the first frame of image data, as specified in Table 2, and subsequently initiates the data acquisition process.

This acquisition process employs either a for or while loop to iterate under predefined conditions (e.g., valid HREF level), sequentially obtaining single-frame image data, where 0xn denotes the n-th frame. After the data is collected through conditional checks and iterative reading, the processor transmits it directly to the NB module. It is essential to ensure that the send buffer size of the USART3 serial port is appropriately configured (e.g., 1 KB) and that the buffer is cleared after each transmission cycle to prepare for subsequent data transfers.

### 3.3. Design of the System Service Layer

The design of the service application layer is implemented using the Spring Boot framework, integrated with Spring Cloud distributed microservice components. Service registration is performed through Eureka. Feign, in conjunction with Nginx, is employed to achieve a two-tier software load balancing mechanism. RabbitMQ serves as the message-oriented middleware. Mycat is utilized as the distributed MySQL middleware. WebSocket technology is applied to establish full-duplex communication with the IoT server. Upon receiving underlying image data, the system performs decoding operations, stores the processed results in a distributed MySQL cluster database, and exposes both RESTful and Web Service APIs. The frontend design adopts the Vue.js framework, deployed within a Node.js runtime environment. Data communication with backend RESTful and Web Service interfaces is implemented using the Axios library, encapsulated within Vue, to execute asynchronous AJAX requests. This architecture supports real-time data retrieval, enabling the visualization of monitoring information directly on the user interface. In cases of anomaly detection, the system issues timely alerts to users to ensure prompt responses.

#### 3.3.1. Design of Image Decoding

Prior to transmitting images from the bottom layer, a data verification header and verification tail are appended, ensuring that the enclosed data conforms to the hexadecimal JPEG image format standard. Before publishing data to the broker via the MQTT protocol, the device must first subscribe to the designated topic [35,36]. MQTT is a TCP-based publish/subscribe protocol known as Message Queuing Telemetry Transport.

After successfully subscribing to the device, the system can continuously receive real-time data transmitted from the underlying layer, thereby establishing WebSocket full-duplex communication with the Spring Boot backend. Upon data reception, the backend first verifies the presence of a data header. If a header is detected, a ‘StringBuffer’ object and an ‘ArrayList’ collection object are instantiated. Once initialized, each incoming data segment is appended to the ‘StringBuffer’ through iterative concatenation. After multiple iterations, the ‘StringBuffer’ contains the complete image data in the required hexadecimal format.

It is important to note that when the underlying image encoding is transmitted to the service layer, the continuous nature of the data stream may cause issues if a ‘String’ type is used for reception. Therefore, the service layer employs a ‘StringBuffer’ object to accommodate variable-length strings. However, improper state management of the ‘StringBuffer’ can lead to residual data from previous transmissions, resulting in concatenation errors. To prevent this, the ‘StringBuffer’ session must be cleared each time new image data is received.

Once a complete image data packet has been assembled, it is extracted and stored in a ‘List’ collection object. Furthermore, during the WebSocket connection between the frontend and backend, variables must be cleared promptly after each transmission cycle to prepare for subsequent data reception. Failure to do so may result in data overwriting, thereby preventing the correct retrieval of hexadecimal-encoded JPEG image data.

After appropriate preprocessing, the data is mapped to the persistence layer via DataJpa, specifically targeting a MySQL database. Once the image data is stored, the hexadecimal representation is converted into a binary format and saved as a file in the project’s root directory, using either the .jpg or .jpeg format. Subsequently, a backend interface is implemented to map the stored data to static image resources. The detailed procedure for decoding JPEG data is illustrated in Figure 7.

#### 3.3.2. NB-IoT Communication Process

The NB module connects to the internet via a SIM card and employs the UDP/IP protocol stack to encapsulate data using the CoAP application-layer protocol for transparent transmission. The encapsulated data is relayed to nearby base stations, enabling communication with remote IoT servers. The IoT servers subsequently forward the image data packets to the service layer for further processing. The specific communication workflow for NB-IoT network residency includes network initialization, communication establishment, command reception, and other steps, as illustrated in Figure 8.

#### 3.3.3. Transparent Transmission Principle Based on CoAP Protocol

NB-IoT primarily employs the lightweight CoAP for transparent data transmission [37]. Terminal devices encapsulate service data in the CoAP format and transmit it to the NB-IoT module. The module issues a GET request method with an optimized set of parameters embedded in the header. Upon receiving the GET request, the network-side service performs resource queries or executes device management operations, subsequently returning a corresponding response to the NB-IoT module. The response header contains the relevant service data, which the NB-IoT module parses and processes before the payload is transparently delivered to the terminal device. This architecture effectively exploits the short-packet characteristics of CoAP over UDP, in conjunction with the power-saving modes of NB-IoT (PSM/eDRX), thereby substantially reducing terminal power consumption and air-interface signaling overhead. The operational flow of transparent transmission within the NB-IoT module is illustrated in Figure 9.

### 3.4. Design of the System Network Application Layer

#### 3.4.1. Database Construction

Currently, mainstream databases include relational systems such as Oracle, MySQL, SQL Server, and SQLite. For database design, this study adopts the MySQL distributed management system. A master-replica cluster is constructed using MySQL middleware to enable read-write separation [38]. Furthermore, the database is vertically partitioned according to business function modules, while horizontal sharding of data is implemented using the modulo (Mod) algorithm. With the aid of MyCat middleware, routing rules and cluster strategies are specified within the configuration (conf) file.

The primary dataset in the proposed design comprises image data. As the image files are stored in hexadecimal JPEG format, the LONGBLOB data type is employed to accommodate this type of content, with the character encoding uniformly set to utf8mb4. Given that the backend development environment is Java-based, the persistence layer framework leverages the Spring ecosystem, specifically the Spring Data JPA framework. This framework automates the generation of Data Definition Language (DDL) table creation statements and facilitates interaction with the MySQL database through the JDBC driver.

#### 3.4.2. Image Processing Algorithms Design

For video content analysis in wide-area surveillance scenarios with minimal background variation, an optimized background subtraction method is often sufficient for effective implementation. This technique fundamentally operates by computing the difference between the current frame and a reference background frame to identify moving objects and associated anomalies [39,40]. The primary implementation challenge lies in constructing an accurate background model, as its precision directly influences the performance of moving object detection. To ensure robust operation, the background model must adapt dynamically to real-world environmental changes through continuous updates, thereby enhancing detection accuracy [41,42,43]. To meet this requirement, our design integrates background subtraction with adaptive background updating for reliable moving object detection. The overall detection workflow is illustrated in Figure 10.

Sensor quantization artifacts result in non-uniform grayscale distributions within background imagery, presenting as both granular and impulse noise. To mitigate these effects, an initial Gaussian smoothing operation employing the canonical 1/16 kernel coefficients is applied prior to subsequent processing stages. This convolution-based filtering technique performs pixel-wise weighted averaging by systematically traversing the kernel across the image plane, with the weighted average computed as follows [44](1)$$C_{kf}\left(x,y\right)=\frac{1}{16}\sum_{i=-1}^{1}\sum_{j=-1}^{1}w\left(i,j\right){\cdot C}_{k}\left(x+i,y+j\right).$$

Per Equation (1), $C_{k}\left(x+i,y+j\right)$ and $\;C_{kf}\left(x,y\right)$, respectively, signify the unfiltered and Gaussian-filtered pixel values at coordinates $\left(x+i,y+j\right)$ in frame *k*. The convolution kernel $w\left(i,j\right)$ assigns weights per spatial position (e.g., center weight $w\left(0,0\right)=4$. Boundary handling employs pixel replication to address incomplete neighborhoods.

An adaptive background update strategy refines $B_{k}\left(x,y\right)$ based on the current frame, as defined in Equation (2), where *α* represents the learning rate. Each updated background pixel constitutes a weighted average between current and historical values. In implementation, *α* = 0.9, and the threshold T = 59.87 is determined through iterative optimization.(2)$$B_{kup}\left(x,y\right)={\alpha B}_{k}\left(x,y\right)+(1-\alpha )C_{kf}\left(x,y\right).$$

Binarization is subsequently performed. The absolute difference map, $D_{k}\left(x,y\right)$ between the current frame and the updated background frame, $B_{kup}\left(x,y\right)$ is computed according to Equation (3).(3)$$D_{k}\left(x,y\right)=\left|C_{kf}\left(x,y\right)-B_{kup}\left(x,y\right)\right|.$$

The foreground region ${Fg}_{k}\left(x,y\right)$ (representing moving objects) was subsequently segmented using threshold T via Equation (4), where T = 59.87 was determined through iterative optimization. The foreground region (representing moving objects) was subsequently segmented using the threshold T, as defined in Equation (4), where T = 59.87 was determined through iterative optimization.(4)$${Fg}_{k}\left(x,y\right)=\left\{\begin{array}{c} 1,\;\;{if(D}_{k}\left(x,y\right)>T),\\ 0,\;\;else.\;\;\;\;\;\;\;\;\;\;\;\;\;\;\;\;\;\;\;\;\;\;\;\; \end{array}\right.$$

Surveillance imagery is represented as a binary image with pixel values {0, 1}, where contours are denoted by 1. Based on the general principle that the contours of moving objects in consecutive frames do not occupy identical positions, further processing is applied: (1) contour pixels retained from previous frames are suppressed (assigned a value of 0) when they are no longer detected, and (2) newly emerged or persistent contours are preserved (assigned a value of 1). This procedure effectively removes static background artifacts while retaining the true contours of moving objects. The system subsequently verifies the persistence of trajectory formation toward image boundaries. Non-persistent moving objects trigger automated alerts, accompanied by the capture of evidentiary imagery.

## 4. System Testing and Analysis 

### 4.1. Image Transmission Success Rate Testing and Analysis 

In the design, the baud rate for the bottom-layer image transmission is configured at 115,200 bps. Since the baud rate is inversely related to the transmission distance, a higher baud rate typically results in a reduced effective communication range. The system test image is shown in Figure 11, and the experimental data are presented in Figure 12. Given the minimal distance between the camera and the main control chip, the transmission success rate achieves 100% at a baud rate of 115,200 bps. The test data reveals a rough positive correlation between the success rate of image transmission and the baud rate, indicating a discrepancy between the theoretical general model (Shannon’s theorem) and specific engineering practices. This is because the communication environment between the camera and the main controller is a highly optimized scenario characterized by high bandwidth and high signal-to-noise ratio. Even with a high baud rate, the actual data rate is significantly lower than the channel capacity, far from reaching the channel limit. Therefore, a high baud rate does not lead to an increase in bit error rate. Instead, the primary source of errors becomes clock desynchronization. As the baud rate increases, the absolute total time required to complete the transmission of a frame of image data decreases, while the absolute clock deviation remains constant. The higher the baud rate, the smaller the accumulated error of clock deviation relative to the entire transmission period, making it less likely to drift beyond the edge of the bit position. Consequently, in this specific application of board-level interconnection, increasing the baud rate emerges as one of the most effective means to reduce transmission errors and enhance the success rate. The test results demonstrate that the system achieves a high success rate in image transmission, which is primarily attributed to the appropriate baud rate configuration that minimizes errors from clock synchronization. This characteristic ensures that basic data transmission remains largely unaffected by variations in illumination conditions.

### 4.2. Server Load Balancing Testing and Analysis

The backend utilizes the Spring Boot framework to develop the web server, integrates Spring Cloud microservices, and employs RabbitMQ as the message queue system. Since images are transmitted concurrently from multiple sources, the server may experience a surge in request traffic, potentially causing a crash. To mitigate this, Feign, integrated within Spring Cloud, is adopted as a load balancer. Regarding the database architecture, MyCat is used as the database middleware, enabling database sharding and read–write splitting through its configuration files. JUC concurrency testing is conducted to compare and analyze the differences in high-concurrency resilience and load performance between Spring Cloud databases and general service loads. The test results are presented in Figure 13. It is important to note that only the backend framework is tested in this context, excluding considerations of thread synchronization and security aspects. Therefore, when incorporating pessimistic and optimistic locking mechanisms into the tests, the success rate of Spring Cloud distributed microservices under high-concurrency request volumes remains at 100% in both scenarios. Tests indicate that the SpringCloud microservices architecture maintains stability under high concurrency conditions. When integrated with load balancers and database sharding strategies, it effectively handles the load from simultaneous multi-location image transmission. This capability is crucial for the system’s stability across diverse operational scenarios.

### 4.3. Testing and Analysis of Accuracy in Identifying Anomalous Images

The system performs threshold segmentation on the JPEG data acquired from the underlying layer, assigning pixel values greater than the threshold to 255 (pure white) and those below the threshold to 0 (pure black). Image anomaly detection is conducted by employing the abnormal image threshold as the classification criterion, which is then compared against inter-frame difference metrics. The data processing success rate was evaluated over 1000 trials, with the test results presented in Figure 14. According to these results, the image recognition accuracy consistently above 98%. These findings demonstrate that the system design is highly reliable. Although the binarization method employed by the system is straightforward, it demonstrates some adaptability to illumination changes, albeit with potential challenges under complex lighting conditions. Future work will explore the introduction of a dynamic threshold adjustment mechanism or the use of complementary multi-sensor data (e.g., from an illuminance sensor) to correct image acquisition, thereby enhancing the system’s overall adaptability to varying illumination.

The system demonstrates considerable robustness to varying illumination in current tests, with its image processing approach proving both reliable and efficient under stable lighting. Concurrently, it exhibits strong adaptability to complex operational scenarios such as high concurrency, where the integrated software-hardware architecture ensures stability during intensive multi-node operations. For applications involving extreme illumination fluctuations or demanding high-fidelity image detail recognition, further optimizations—such as introducing adaptive thresholds and refining node collaboration mechanisms—will be required.

## 5. Conclusions

The system architecture guarantees reliable and stable performance throughout the entire workflow: starting from data acquisition via communication between the underlying ARM processor and the OV7725 camera sensor, continuing through image processing performed by the STM32F407 master control chip, and extending to server-side operations including decoding of image data transmitted to the user application server, MySQL database storage, and image format conversion. The final processing stages encompass image binarization, user notification for abnormal images, and comprehensive data visualization. Experimental tests have proven that the system’s transmission success rate has reached 100%. When the server faces high concurrent request volumes, the success rate can also stably maintain at 100%. Although the success rate of abnormal image recognition can be maintained above 98%, it has not yet reached 100% reliability, indicate that there is still room for improvement in our image processing algorithms.

By leveraging state-of-the-art Narrow Band Internet of Things (NB-IoT) technology, the design facilitates outdoor wide-area monitoring and early warning functionalities integrated with data visualization. This approach effectively overcomes the inherent limitations of traditional wide-area solutions—namely excessive bandwidth consumption, high power requirements, and constrained device connectivity—thereby achieving extensive coverage, ultra-low power consumption, and massive device connectivity for networked monitoring and visualized data operations. Regarding the research direction of wide-area visual monitoring for narrowband Internet of Things (NB-IoT), our future work will focus on utilizing NB-IoT to transmit more efficient image formats, enhancing the durability of the system across various lighting conditions and operational scenarios, and improving the accuracy and timeliness of abnormal image detection.

## Figures and Tables

**Figure 1 sensors-25-06589-f001:**
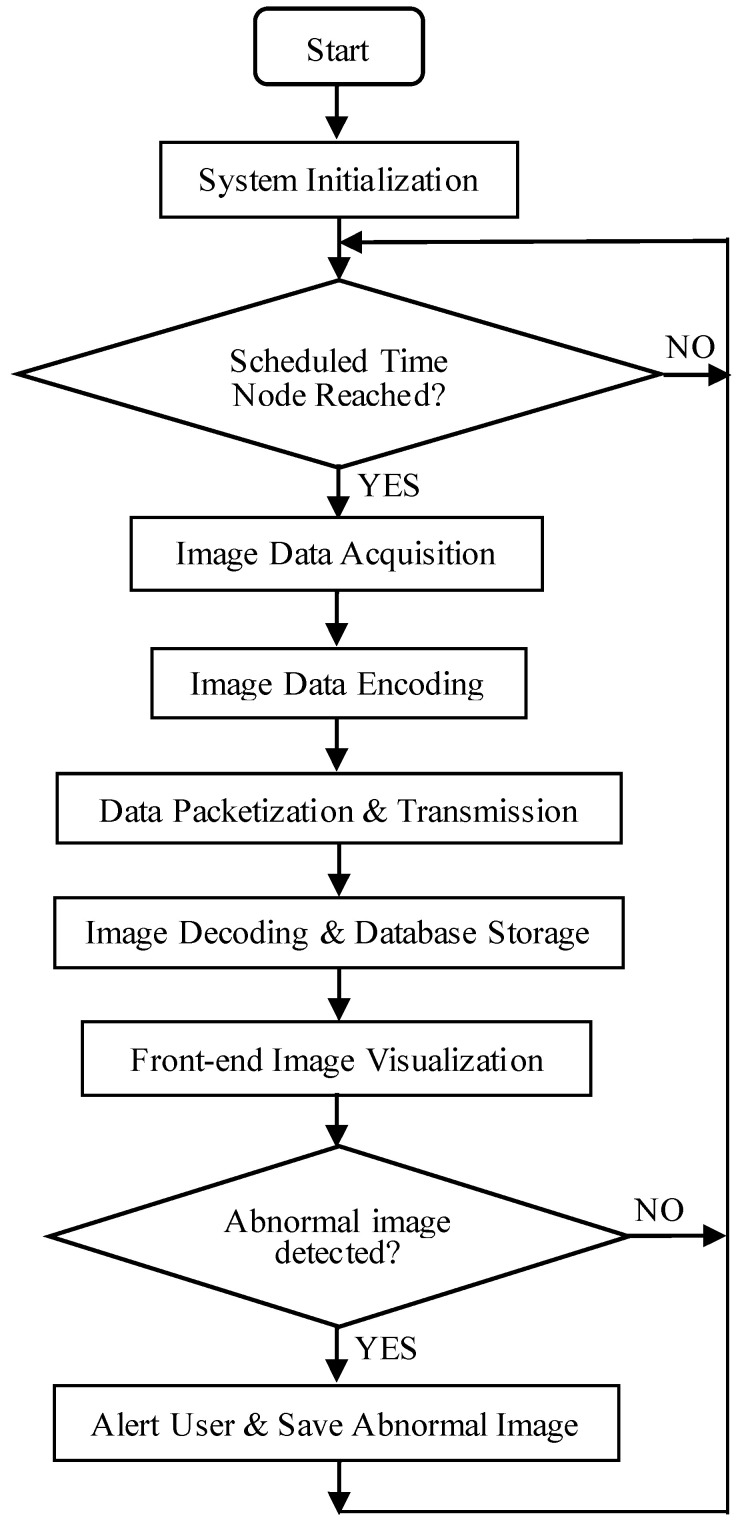
Flowchart of the overall architecture of the system program.

**Figure 2 sensors-25-06589-f002:**
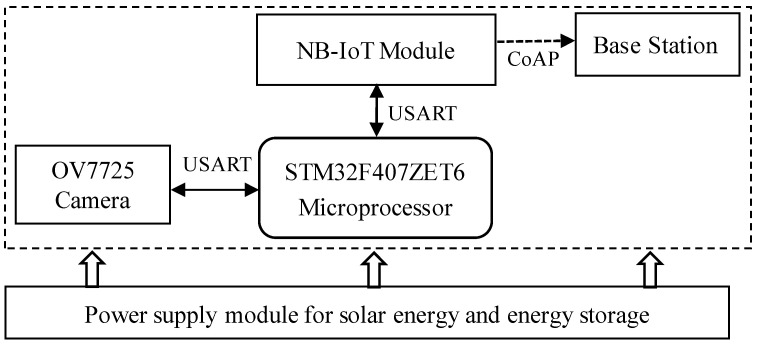
System hardware architecture diagram.

**Figure 5 sensors-25-06589-f005:**
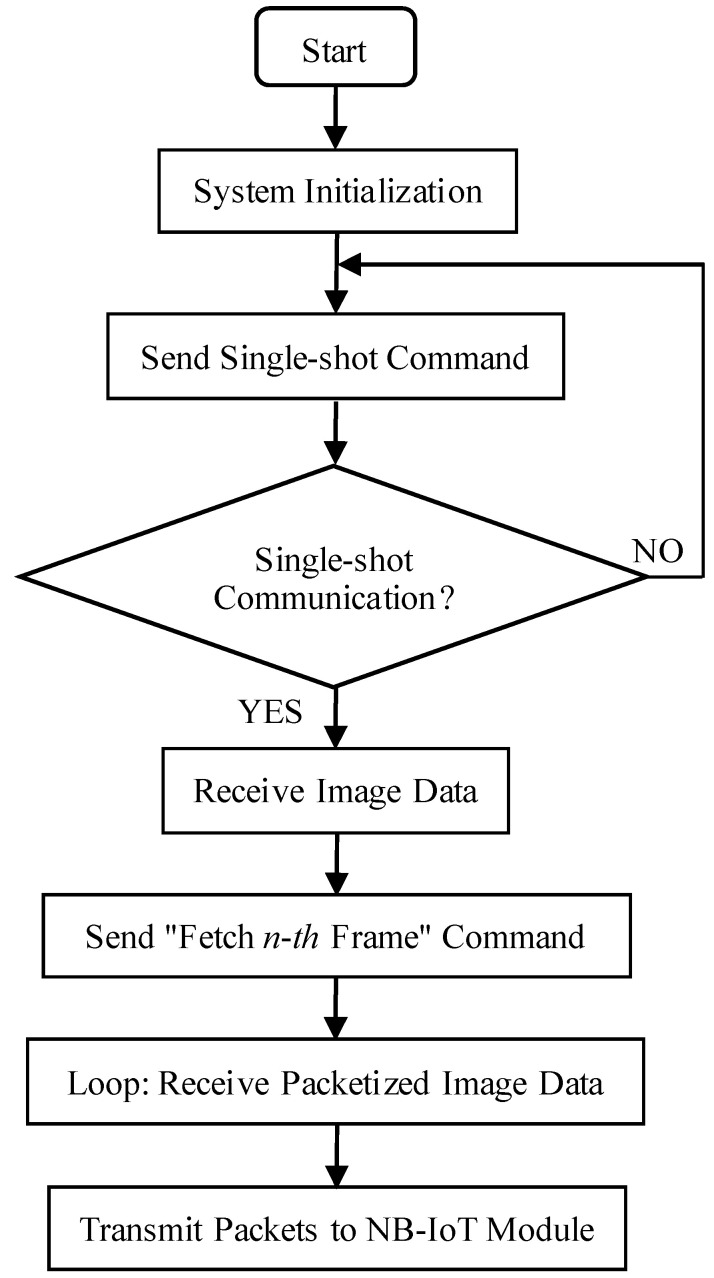
Master UART communication flowchart.

**Figure 6 sensors-25-06589-f006:**
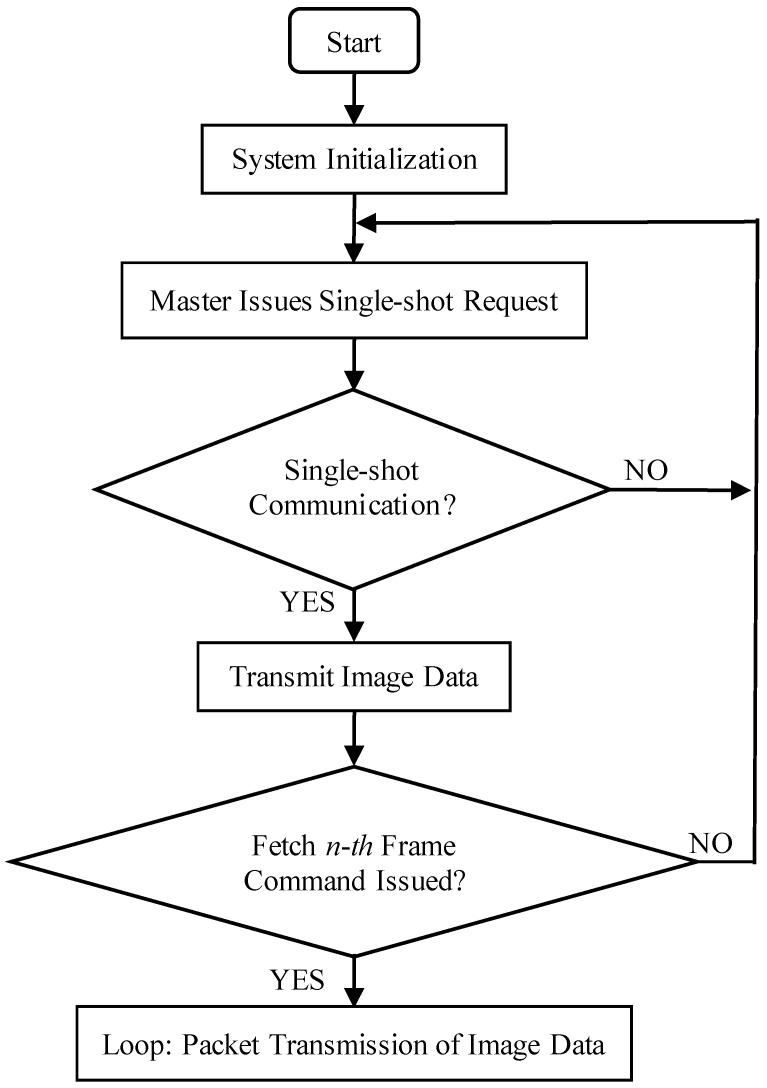
OV7725 communication flowchart.

**Figure 7 sensors-25-06589-f007:**
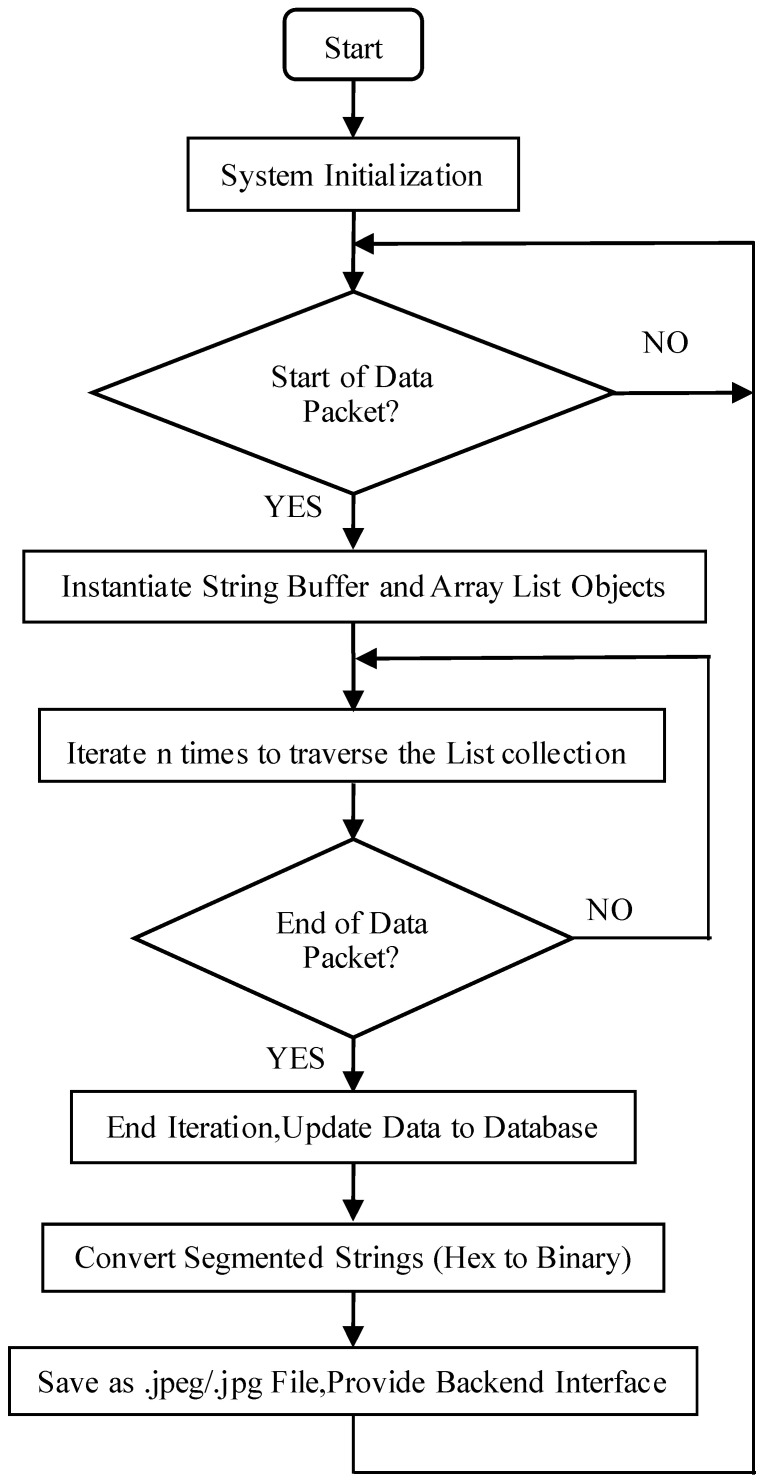
JPEG image data decoding workflow.

**Figure 8 sensors-25-06589-f008:**
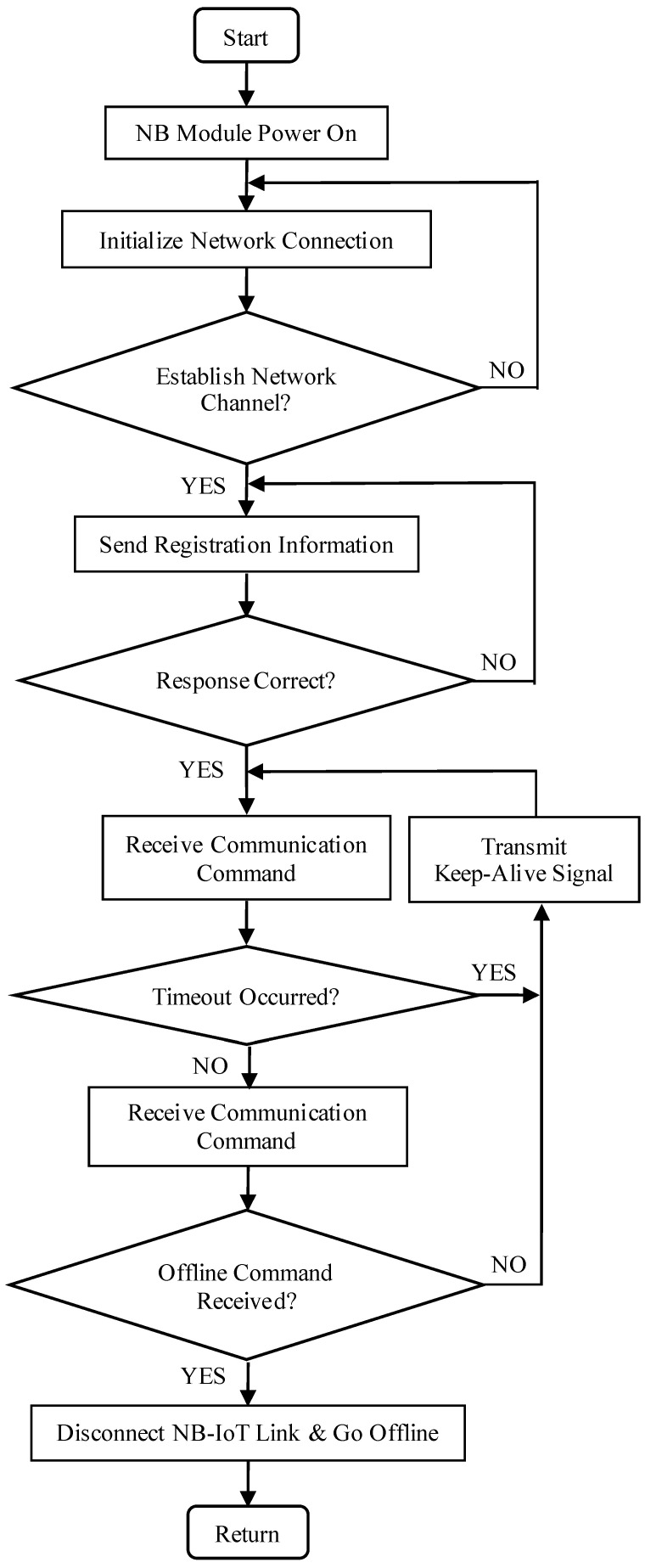
NB-IoT-to-server communication protocol flow.

**Figure 9 sensors-25-06589-f009:**
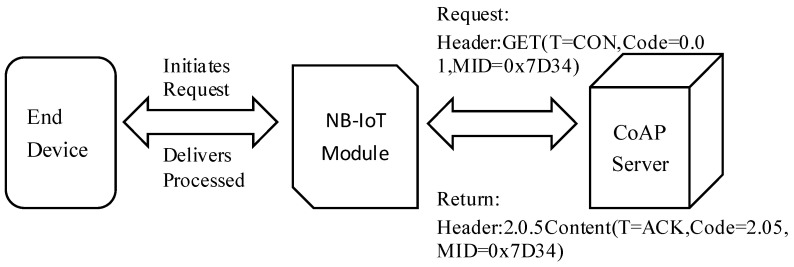
CoAP-based data passthrough mechanism diagram.

**Figure 10 sensors-25-06589-f010:**
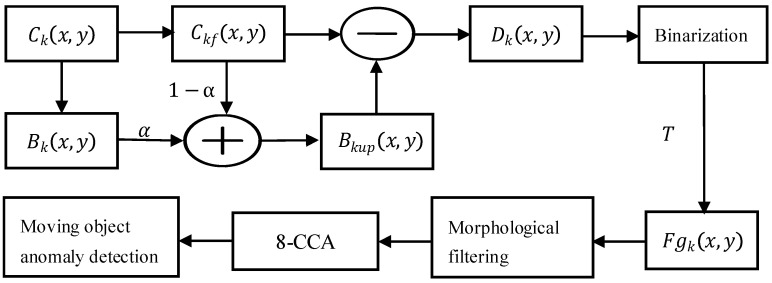
Processing pipeline of optimized background subtraction.

**Figure 3 sensors-25-06589-f003:**
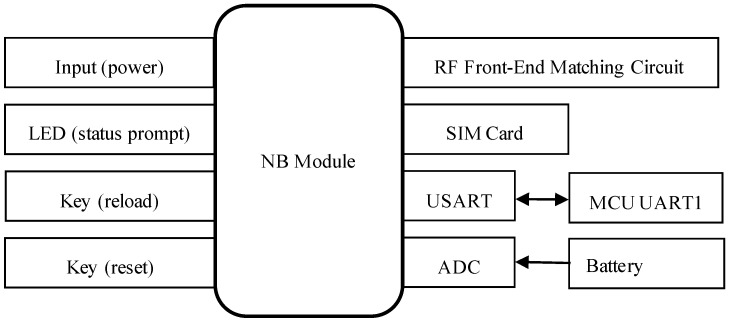
Design diagram of the peripheral circuit of the NB module.

**Figure 4 sensors-25-06589-f004:**
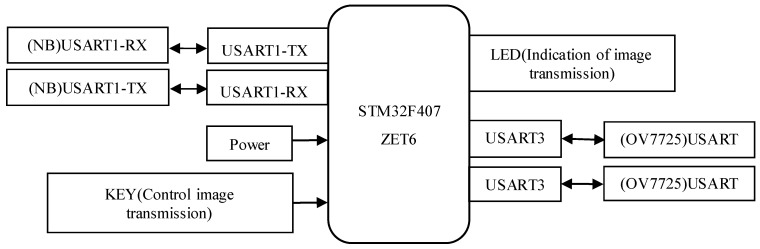
Design diagram of the peripheral circuit for the main control STM32F407ZET6.

**Figure 11 sensors-25-06589-f011:**
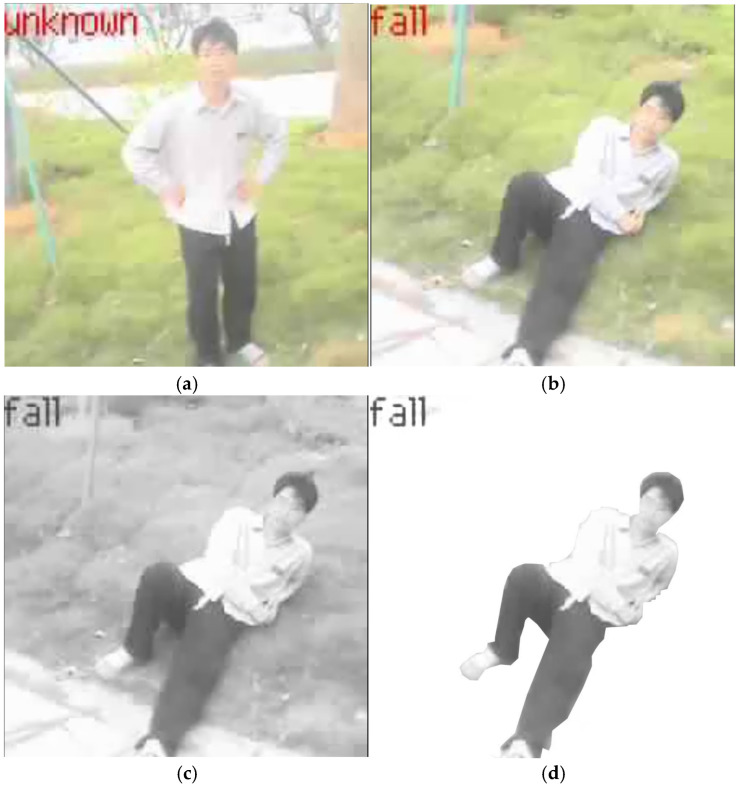
System experiment images.(**a**) Normal walking image on site. (**b**) Abnormal walking image on site. (**c**) Abnormal walking grayscale image. (**d**) Abnormal walking grayscale image with background removed.

**Figure 12 sensors-25-06589-f012:**
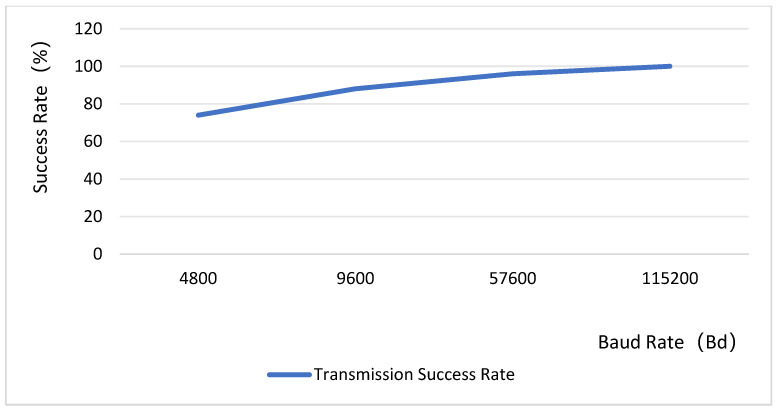
System image transmission success rate test.

**Figure 13 sensors-25-06589-f013:**
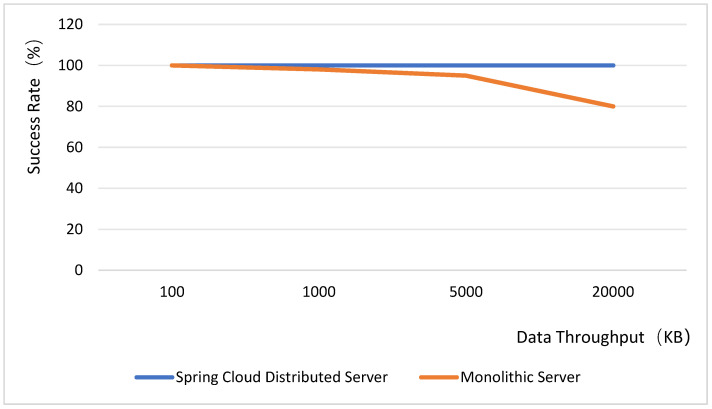
Server concurrent load capacity test.

**Figure 14 sensors-25-06589-f014:**
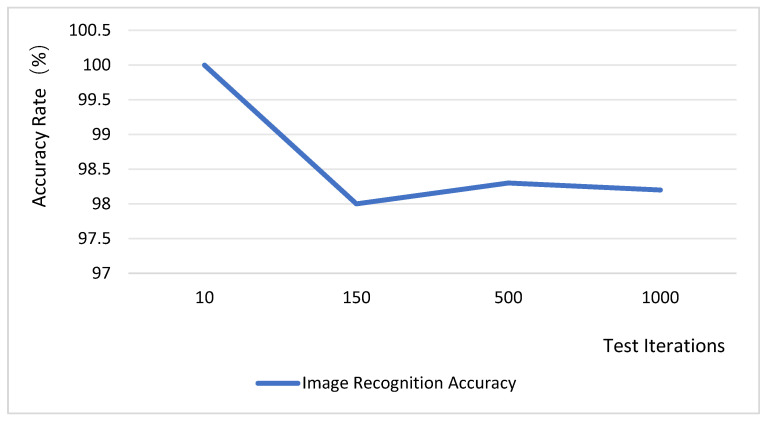
Image recognition accuracy rate test.

**Table 1 sensors-25-06589-t001:** Relevant indicator information of communication module.

Category	Infrared	Bluetooth	Wifi	ZigBee	GPRS	LoRaWAN
Transmission Range	5 m	10 m	100 m	300 m	300 m	15 km
Maximum Power Consumption	10 mW	100 mW	100 mW	150 mW	2 w	50 mW
Transmission Rate	100 kbps	1–2 Mbps	300 Mbps	250 kbps	20 kbps	0.3–62.5 kbps

**Table 2 sensors-25-06589-t002:** OV7725 camera sensor communication protocol.

Hex-Format Command	Function Code
0x55 0x48 0x00 0x31 0x00 0x02 0x23	Single-shot Command, Formatted Image Data Response
0x55 0x45 0x00 0x31 0x01 0x00 0x23	Fetch First-frame Hex Data
0x55 0x45 0x00 0x31 0xn 0x00 0x23	Fetch n-frame Hex Data

## Data Availability

The original contributions presented in the study are included in the article; further inquiries can be directed to the corresponding authors.

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
