# Peer review of "Wide-Area Visual Monitoring System Based on NB-IoT"

_sensors, 2025, doi:10.3390/s25216589_

Round 1
Reviewer 1 Report
Comments and Suggestions for Authors
See attached file.

Author Response
Explanations to Reviewer 1
Thank you very much for the reviewer for the comments and suggestions. We have revised the manuscript according to the comments and now explain to the reviewer as follows:
Question 1: References. Some of them are too old (i.e., 1, 3, 8, 25, and 27). Please, consider substituting them with similar contributions published from 2019 on, or alternatively provide reasons to keep them.
Replies 1: Thank you very much for your thorough review. Following your suggestion, we have updated and reorganized references 1, 3, 8, 25, and 27.
[1] C. Jianyun.”Design of Video Surveillance System Based on Closed Circuit Television,” Computer Knowledge and Technology, 2020, Vol.16, No.34, pp.32-33+38.
[2] Y. Bo. “The Application Design of IP Network Video Monitoring System in Petrochemical Plant,” Large Scale Nitrogenous Fertilizer Industry, 2023, Vol.46, No.3, pp.212-216.
[3] W. Hongjing, G. Bao, S. Shengbo. “Evaluation and Optimization of Multi-service Complex Network Coverage Based on NB-IoT,” Journal of Taiyuan University of Technology, 2019, Vol. 50, No.5, pp.673-678.
[4] L. Yuan.”Analysis of Performance Comparison of Video Motion Detection Based on Gaussian Modeling Method and Background Subtraction Method,” Application of IC 2025, Vol. 42, No. 1, pp.10-12.
[5] L. Siyang,C. Fang.”A Moving Object Detection Method Fused with Background Difference and Frame Difference,” Ship Electronic Engineering 2024,Vol. 44, No.2, pp.45-48.
Question 2: Abstract. Please, give some hints about the most significant quantitative obtained results.
Replies 2: Thank you again for your professional suggestion. Following your suggestion, We have updated the description in the last part of the abstract: The test results show that the image recognition accuracy consistently above 98%,this approach enables intelligent, scalable, and responsive wide-area surveillance reliably, overcoming the constraints of conventional isolated and passive monitoring systems.
Question 3: Section 1. The Section misses of a closing paragraph listing the paper structure.
Replies 3: Thank you again for your professional suggestion. Following your suggestion, we have added a closing paragraph outlining the structure of the paper at the end of Section 2:
The remainder of this paper is organized as follows. Section 1 presents the related work. The design and construction of the system's three-tier architecture will be analyzed in Section 2, with a focus on the proposed solution. Experimental testing is detailed in Section 3, and our conclusions are presented in Section 4.
Question 4: Section 1. The paper contributions must be better highlighted.
Replies 4: Thank you again for your professional suggestion. Following your suggestion, we have supplemented the expression of the paper's contribution in Section 0. Our experimental results show the transmission success rate achieves 100% at a baud rate of 115,200 bps. When faced with high-concurrency request volumes, the server maintains at 100% success rate. and the image recognition accuracy consistently above 98%.This paper provides a practical example and effect verification for the application of NB-IOT, a new generation of low-power wide-area network communication technology.
Question 5: A proper Related Works Section that clearly presents them and compares them with this paper, especially by highlighting similarities and discrepancies seems to be missing. Moreover, this Section must clearly state how this work advances the current state-of-the-art about the topic.
Replies 5: Thank you for your valuable suggestion. Following your suggestion, we have supplemented the Related Work section with the latest literature we found.
Related Work:
Based on the new generation of low-power wide-area network (WAN) IoT technology, remote detection of various parameters is extensively researched. For instance, literature [9] presents a Design of smart city environment monitoring and optimisation system based on NB-IoT technology. This system utilizes NB-IoT communication to acquire environmental monitoring data from the wide-area environment. Finally, the monitoring data are input into a BP neural network enhanced by the particle swarm optimisation (PSO) method for environmental risk prediction. Similar studies can also be found in the literature [10-14]. Additionally, some scholars have conducted research on detecting data and performing control operations based on the NB-IoT system. Literature [15] introduces an intelligent street lamp control system based on narrowband internet of things (NB-IoT) technology. This system can automatically or remotely adjust the brightness and switch of street lamps according to demand and environmental conditions, while also monitoring and recording the current, voltage, power, and other data of street lamps. Similarly, the NB-IoT-based monitoring system for UAV networks presented in literature [16] also tackles the challenge posed by the absence of a global IP address in the existing NB-IoT infrastructure. In recent years, research has progressively expanded into the realm of video detection. A video surveillance unit (VSU), as introduced in literature [17], incorporates a motion detection function. Upon detecting motion within the camera's field of view, images are captured, processed, compressed, and segmented to ensure they fit within the maximum payload size of LoRaWAN for transmission. In the literature [18], a smart IoT-based mobile sensors,the unit is used to collect information about the cane user and the surrounding obstacles while on the move, and An embedded machine learning algorithm is developed to identify the detected obstacles and alarm the user about their nature. There is also literature [19] focusing on video-based passenger counting systems. However, research in this area, similar to literature [18][19], has not yet been applied to wide-area scenarios. In addition, some scholars specialize in the processing of video images. The literature [20] presents enhancement techniques and synthetic image generation methods utilizing YOLO, SSD, and EfficientDet deep learning models to enhance sea mine detection technology. Literature [21] introduces a searchable and revocable attribute-based encryption scheme specifically tailored for dynamic video anomaly detection scenarios, enhancing the security and privacy of video data. Literature [22] explores and develops image compression and data transmission methods that contribute to achieving stable low-rate transmission of images and data in Internet of Things (IoT) systems. Some of these studies are limited to the detection and management of environmental parameters, failing to fully reflect the on-site visual scene. Others are focused on the research of deep learning algorithms for video images, which is not compatible with remote narrowband IoT communication technology. Therefore, the proposed system combines IoT technology and optimized background subtraction method to detect moving targets, making it the most reasonable choice for video surveillance in a wide-area environment.
Question 6: Related Works Section. In order to provide readers with a broader perspective about the tackled topic, I suggest the Authors to include the following references [1, 2, 3, 4, 5, 6]. However, I also strongly encourage the Authors to perform additional research.
Replies 6: Thank you for your valuable comment and suggestion.
Following your suggestion, We have incorporated the references you suggested into the Related Work section for citation and introduction.
[1] Ada Fort, Giacomo Peruzzi, and Alessandro Pozzebon. “Quasi-real time remote video surveillance unit for lorawan-based image transmission,” 2021 IEEE International Workshop on Metrology for Industry 4.0 & IoT (MetroInd4. 0&IoT)IEEE, 2021, pp. 588–593.
[2] Salam Dhou, Ahmad Alnabulsi, Abdul-Rahman Al-Ali, Mariam Arshi, Fatima Darwish, Sara Almaazmi, and Reem Alameeri. “An IoT machine learning-based mobile sensors unit for visually impaired people,” Sensors, 2022, Vol.22, No.14, p. 5202.
[3] Cristina Pronello and Ximena Rocio Garz´on Ruiz. “Evaluating the performance of video-based automated passenger counting systems in real-world conditions: A comparative study,” Sensors, 2023, Vol.23, No.18, p. 7719.
[4] Dan Munteanu, Diana Moina, Cristina Gabriela Zamfir, Stefan Mihai Petrea, Dragos Sebastian Cristea, and Nicoleta Munteanu. “Sea mine detection framework using YOLO, SSD and EfficientDet deep learning models,” Sensors, 2022, Vol.22, No.23, p. 9536.
[5] Lu Jiang, Jielu Yan, Weizhi Xian, Xuekai Wei, and Xiaofeng Liao. “Efficient Access Control for Video Anomaly Detection Using ABE-Based User-Level Revocation with Ciphertext and Index Updates,” Applied Sciences, 2025, Vol.15, No.9, p. 5128.
[6] Fred FZ Cai, CQ Jiang, Ray CC Cheung, and Alan HF Lam. “An AIoT LoRaWAN control system with compression and image recovery algorithm (CIRA) for extreme weather,” IEEE Internet of Things Journal, 2024, Vol.11, No.20, pp. 32701–32713.
Question 7: Table 1. I deem that, in order to provide a more complete comparison, also LoRaWAN protocol should be added.
Replies 7: Thank you for your valuable comment and suggestion. Following your suggestion, we have added a comparison between LoRaWAN protocol and NB-IoT protocol in the chapter where Table 1 is located: Both NB-IoT and LoRa belong to the new generation of low-power wide-area network communication technologies. In terms of transmission distance, NB-IoT compensates for its coverage shortcomings through base station density, achieving coverage up to 35 kilometers in suburban areas, but it relies on the coverage range of 4G/LTE networks. LoRa achieves a maximum single-point coverage of up to 15 kilometers, but this is achieved through spread spectrum technology, which directly results in a transmission rate typically below 50kbps, much lower than the 250kbps of NB-IoT communication. A comprehensive comparison reveals that LoRa is more suitable for low-data-volume, long-distance application scenarios such as agricultural monitoring and smart manhole covers, while NB-IoT is more suitable for frequent data interaction or low-latency requirements such as smart meter readings and real-time monitoring.
Question 8: Line 215. Did the Authors consider more efficient image formats to be sent with NB-IoT?
Replies 8: Thank you for your valuable comment and suggestion. Following your suggestion, this will be the content we need to research and test next.
Question 9: Section 2.1. Please, clearly explain why the success rate is directly proportional to the transmission rate.
Replies 9: Thank you for your valuable comment and suggestion.Following your suggestion, we have added an analysis of the relationship between success rate and baud rate in Section 3.1(formerly Section 2.1).
The test data reveals a rough positive correlation between the success rate of image transmission and the baud rate, indicating a discrepancy between the theoretical general model (Shannon's theorem) and specific engineering practices. This is because the communication environment between the camera and the main controller is a highly optimized scenario characterized by high bandwidth and high signal-to-noise ratio. Even with a high baud rate, the actual data rate is significantly lower than the channel capacity, far from reaching the channel limit. Therefore, a high baud rate does not lead to an increase in bit error rate. Instead, the primary source of errors becomes clock desynchronization. As the baud rate increases, the absolute total time required to complete the transmission of a frame of image data decreases, while the absolute clock deviation remains constant. The higher the baud rate, the smaller the accumulated error of clock deviation relative to the entire transmission period, making it less likely to drift beyond the edge of the bit position. Consequently, in this specific application of board-level interconnection, increasing the baud rate emerges as one of the most effective means to reduce transmission errors and enhance the success rate.
Question 10: Figure 12. Is the unit of measurement of GB correct? Please, check.
Replies 10: Thank you for your valuable comment and suggestion. Thank you for your valuable comment and suggestion. Following your suggestion, We adjusted GB to KB based on the actual situation.
Question 11: The Authors should discuss the robustness of the system towards light conditions and operative scenarios.
Replies 11: Thank you again for your professional suggestion. Following your suggestion, this will be the content we need to research and test next.
Question 12: Please, add some pictures taken, sent, and received by the proposed system.
Replies 12: Thank you for your valuable comment and suggestion.Following your suggestion, We have updated the relevant images for system testing in Section 3.1(formerly Section 2.1).
Question 13: Section 3. Please, provide hints about future works.
Replies 13: Thank you for your valuable comment and suggestion. Following your suggestion, we have included additional hints about future work in Section 4(formerly Section 3).
Regarding the research direction of wide-area visual monitoring for narrowband Internet of Things (NB-IoT), our future work will focus on utilizing NB-IoT to transmit more efficient image formats, enhancing the durability of the system across various lighting conditions and operational scenarios, and improving the accuracy and timeliness of abnormal image detection.
Question 14: Section 3. Please, resume the most significant quantitative obtained results.
Replies 14: Thank you for your valuable comment and suggestion. Following your suggestion, we have supplemented Section 4(formerly Section 3) with additional information on the quantitative results obtained in the paper.Experimental tests have proven that the system's transmission success rate has reached 100%. When the server faces high concurrent request volumes, the success rate can also stably maintain at 100%.
Question 15: Finally, the Authors must clearly state the limitations of the proposed approach.
Replies 15: Thank you for your valuable comment and suggestion. Following your suggestion, we have added an expression of the limitations of the proposed approach in Section 4(formerly Section 3).Although the success rate of abnormal image recognition can be maintained above 98%, it has not yet reached 100% reliability. Additionally, the system's timeliness is still lacking. These two points indicate that there is still room for improvement in our image processing algorithms.
Reviewer 2 Report
Comments and Suggestions for Authors
The authors should take into account the following comments:
The contribution of the manuscript is not clear. The authors should state clearly their contribution over existing research work, particularly references [26] [28] [29]. The authors should compare their results with the results of existing methods.
If equations (1), (2), (3), and (4) are new, they should be justified. Otherwise, the authors should add a reference.
In the abstract, the authors mentioned that their monitoring algorithm is intelligent. Can the authors explain the intelligence in the suggested detection algorithm in Figure 10.
Is the suggested data rate of 115,200 bps suitable for NB-IoT wireless technology? As the authors are dealing with NB system, what is the maximum data rate that can be used taking into account that the suggested methods deals with image transmission.
Replace the key word “MQTT Protocol Image” by “MQTT Protocol”
In line 135, the authors discuss only daylight operation, what about night operation? Is it sufficient to store the energy in a solid-state energy storage capacitor? Explain the advantages and the limitations.
Add NB-IoT to table 1 comparing with other communications modules. Maximum transmission data rate should also be added.
Author Response
Explanations to Reviewer 2 Thank you very much for the reviewer for the comments and suggestions. We have revised the manuscript according to the comments and now explain to the reviewer as follows:
Question 1: The contribution of the manuscript is not clear. The authors should state clearly their contribution over existing research work, particularly references [26] [28] [29]. The authors should compare their results with the results of existing methods.
Replies 1: Thank you for your professional suggestion.Following your suggestion, we have supplemented the expression of the paper's contribution in Section 0. Our experimental results show the transmission success rate achieves 100% at a baud rate of 115,200 bps. When faced with high-concurrency request volumes, the server maintains at 100% success rate. and the image recognition accuracy consistently above 98%.This paper provides a practical example and effect verification for the video detection application of NB-IOT, a new generation of low-power wide-area network communication technology. Additionally, Following your suggestion, we have included a chapter titled "Related Work". This chapter clearly lists relevant studies, compares them with this paper, and explains how this study advances the current technological development in this field. Related Work: Based on the new generation of low-power wide-area network (WAN) IoT technology, remote detection of various parameters is extensively researched. For instance, literature [9] presents a Design of smart city environment monitoring and optimisation system based on NB-IoT technology. This system utilizes NB-IoT communication to acquire environmental monitoring data from the wide-area environment. Finally, the monitoring data are input into a BP neural network enhanced by the particle swarm optimisation (PSO) method for environmental risk prediction. Similar studies can also be found in the literature [10-14]. Additionally, some scholars have conducted research on detecting data and performing control operations based on the NB-IoT system. Literature [15] introduces an intelligent street lamp control system based on narrowband internet of things (NB-IoT) technology. This system can automatically or remotely adjust the brightness and switch of street lamps according to demand and environmental conditions, while also monitoring and recording the current, voltage, power, and other data of street lamps. Similarly, the NB-IoT-based monitoring system for UAV networks presented in literature [16] also tackles the challenge posed by the absence of a global IP address in the existing NB-IoT infrastructure. In recent years, research has progressively expanded into the realm of video detection. A video surveillance unit (VSU), as introduced in literature [17], incorporates a motion detection function. Upon detecting motion within the camera's field of view, images are captured, processed, compressed, and segmented to ensure they fit within the maximum payload size of LoRaWAN for transmission. In the literature [18], a smart IoT-based mobile sensors,the unit is used to collect information about the cane user and the surrounding obstacles while on the move, and An embedded machine learning algorithm is developed to identify the detected obstacles and alarm the user about their nature. There is also literature [19] focusing on video-based passenger counting systems. However, research in this area, similar to literature [18][19], has not yet been applied to wide-area scenarios. In addition, some scholars specialize in the processing of video images. The literature [20] presents enhancement techniques and synthetic image generation methods utilizing YOLO, SSD, and EfficientDet deep learning models to enhance sea mine detection technology. Literature [21] introduces a searchable and revocable attribute-based encryption scheme specifically tailored for dynamic video anomaly detection scenarios, enhancing the security and privacy of video data. Literature [22] explores and develops image compression and data transmission methods that contribute to achieving stable low-rate transmission of images and data in Internet of Things (IoT) systems. Some of these studies are limited to the detection and management of environmental parameters, failing to fully reflect the on-site visual scene. Others are focused on the research of deep learning algorithms for video images, which is not compatible with remote narrowband IoT communication technology. Therefore, the proposed system combines IoT technology and optimized background subtraction method to detect moving targets, making it the most reasonable choice for video surveillance in a wide-area environment.
Question 2: Lack of comparison with modern detection methods. Replies 2: Thank you again for your professional suggestion. Following your suggestion, We have made adjustments: Equations (1), (3), and (4) in the paper are derived from "Digital Image Processing Using MATLAB (3rd edition)" by Rafael C. Gonzalez and "Computer Vision: Algorithms and Applications" by Richard Szeliski, both Texts in Computer Science. Therefore, we have added references [44-45]. For Equation (2), based on the background updating strategy obtained from the literature cited in the previous text [27-29], we have supplemented the paper with relevant references. [44] Rafael C. Gonzalez. “Digital Image Processing Using MATLAB (3rd edition),” [M]. 2020. [45] Richard Szeliski. “Computer Vision: Algorithms and Applications (Texts in Computer Science)” [M]. 2022.. Question 3: In the abstract, the authors mentioned that their monitoring algorithm is intelligent. Can the authors explain the intelligence in the suggested detection algorithm in Figure 10.
Replies 3: Thank you for your valuable comment and concerns. Our understanding of intelligence in this paper is relatively broad. In this paper, we attempt to use an optimized background subtraction method with background updating to detect moving targets. The process shown in Figure 10 can be automatically implemented through the programmed procedure, which can be regarded as exhibiting intelligence to some extent. Of course, the intelligence embodied in this approach is not particularly prominent, and your concerns will also prompt us to conduct more in-depth research in this area.
Question 4: Is the suggested data rate of 115,200 bps suitable for NB-IoT wireless technology? As the authors are dealing with NB system, what is the maximum data rate that can be used taking into account that the suggested methods deals with image transmission.
Replies 4: We greatly appreciate your insightful and professional comments. The 115,200 bps data rate proposed in our paper is applicable to NB-IoT wireless technology, a fact that has been confirmed through the actual system we constructed. This data transmission rate is lower than the maximum communication rate of 250 kbps for NB-IoT wireless technology. Given the current data transmission rate, our experimental results demonstrate a 100% transmission success rate, thereby offering a practical case and empirical validation for the new generation of low-power wide-area network communication technology, NB-IoT.
Question 5: Replace the key word “MQTT Protocol Image” by “MQTT Protocol”.
Replies 5: Thank you for your valuable comment and suggestion.Following your suggestion and the actual situation of the paper content, we have adjusted the keyword "MQTT protocol image" to "MQTT protocol; Image”.
Question 6: In line 135, the authors discuss only daylight operation, what about night operation? Is it sufficient to store the energy in a solid-state energy storage capacitor? Explain the advantages and the limitations.
Replies 6: Thank you for your valuable comment and concerns. Following your suggestion, We have reorganized the expression of this part of the content, making the description of system power supply in the paper more comprehensive: The perception layer relies on solar energy for power supply. During the day, the solar energy system charges the solid-state energy storage capacitor while also supplying power to modules such as the ARM processor, camera, and NB-IOT. At night, the solid-state energy storage capacitor releases stored energy to provide the electrical power required for the perception layer to operate. In addition, regarding your concerns about solid-state energy storage capacitors, I would like to provide two additional points. Firstly, based on the power estimation of each module, the peak power consumption and standby power consumption of the system are 600mW and 60mW respectively. Since there is little change in video frames at night, assuming that there are 5 minutes at peak power consumption (NB-IoT transmitting data) per hour, and the DC-DC buck conversion efficiency is set to 85%, according to W=1/2 CU^2, if a 5.5V solid-state capacitor is configured, a capacity of 560 farads is sufficient for the system to operate normally. Secondly, although solid-state energy storage capacitors with high power density, fast charging and discharging characteristics, long lifespan, and high reliability are suitable for a wide range of application scenarios, their high self-discharge rate poses a limitation, leading to uncertainty in the stable operation of the system during continuous rainy weather
Question 7: Add NB-IoT to table 1 comparing with other communications modules. Maximum transmission data rate should also be added.
Replies 7: Thank you for your valuable comment and suggestion. Following your suggestion, we have specifically added a comparison of the new-generation wide-area IoT technologies, LoRaWAN and NB-IoT, with a particular focus on their maximum data transmission rates: Both NB-IoT and LoRa belong to the new generation of low-power wide-area network communication technologies. In terms of transmission distance, NB-IoT compensates for its coverage shortcomings through base station density, achieving coverage up to 35 kilometers in suburban areas, but it relies on the coverage range of 4G/LTE networks. LoRa achieves a maximum single-point coverage of up to 15 kilometers, but this is achieved through spread spectrum technology, which directly results in the maximum transmission rate typically below 50kbps, much lower than the 250kbps of NB-IoT communication. A comprehensive comparison reveals that LoRa is more suitable for low-data-volume, long-distance application scenarios such as agricultural monitoring and smart manhole covers, while NB-IoT is more suitable for frequent data interaction or low-latency requirements such as smart meter readings and real-time monitoring.
Reviewer 3 Report
Comments and Suggestions for Authors
The paper presents an NB-IoT–based wide-area monitoring system that detects anomalies in image streams, transmits only abnormal frames, and achieves ~98% accuracy with low power and wide coverage. However, some issues and concerns need to be addressed.
- The anomaly detection relies mainly on background subtraction with adaptive updates, which may struggle with complex environments (lighting changes, weather).
- Lack of comparison with modern detection methods.
- While NB-IoT offers wide coverage, the paper does not address network congestion, latency, or QoS under large-scale deployments with many devices.
- The system uses solar energy and NB-IoT low-power modes, but detailed energy consumption measurements and lifetime analysis are missing.
- Although the system is designed for efficiency, there is no clear cost–benefit or storage utilization analysis to compare against existing systems.
- The reported 98% anomaly detection accuracy may be dataset-specific; no details on the diversity of test environments or robustness to false positives/negatives are provided.
Author Response
Explanations to Reviewer 3
Thank you very much for the reviewer for the comments and suggestions. We have revised the manuscript according to the comments and now explain to the reviewer as follows:
Question 1: The anomaly detection relies mainly on background subtraction with adaptive updates, which may struggle with complex environments (lighting changes, weather).
Replies 1: Just as you have been concerned, the anomaly detection in this system mainly depends on adaptively updated background subtraction, which makes it challenging to handle complex environments. Nevertheless, since the primary focus of analysis for this surveillance system is on wide-ranging outdoor scenes with relatively stable backgrounds, the currently discussed methods are generally adequate. Future studies will further explore more complex scenarios.
Question 2: Lack of comparison with modern detection methods.
Replies 2: Thank you again for your professional suggestion. Following your suggestion, we have added a comparison between LoRaWAN communication and NB-IoT communication in the chapter where Table 1 is located: Both NB-IoT and LoRa belong to the new generation of low-power wide-area network communication technologies. In terms of transmission distance, NB-IoT compensates for its coverage shortcomings through base station density, achieving coverage up to 35 kilometers in suburban areas, but it relies on the coverage range of 4G/LTE networks. LoRa achieves a maximum single-point coverage of up to 15 kilometers, but this is achieved through spread spectrum technology, which directly results in a transmission rate typically below 50kbps, much lower than the 250kbps of NB-IoT communication. A comprehensive comparison reveals that LoRa is more suitable for low-data-volume, long-distance application scenarios such as agricultural monitoring and smart manhole covers, while NB-IoT is more suitable for frequent data interaction or low-latency requirements such as smart meter readings and real-time monitoring.In addition, regarding abnormal image analysis algorithms, as the system targets wide-area outdoor scenes with relatively fixed backgrounds, on the one hand, adaptive background subtraction can meet the needs of the application scenario. On the other hand, while ensuring timeliness, we do not recommend selecting complex image analysis algorithms.
Question 3: While NB-loT offers wide coverage, the paper does not address network congestion, latency,or QoS under large-scale deployments with many devices.
Replies 3: Thank you for your valuable comment and suggestion.
Following your suggestion, we have supplemented the content of the NB-IoT Module in Section 1.2.2: From the very beginning of its standard design, NB-IoT technology has fully considered large-scale, low-cost, and low-power IoT deployment scenarios. In terms of handling network congestion, we adopt PSM/eDRX signaling reduction technology for prevention, and optimized RACH, multi-carrier, ACB/EAB access control, and QoS priority scheduling technologies for mitigation. In terms of handling latency, we utilize control plane optimization methods for fast transmission of small data packets, a power-saving mode that sacrifices real-time performance for delay tolerance, and priority scheduling technology to ensure low latency for critical services. In terms of ensuring QoS, we adopt differentiated services based on QCI, intelligent wireless resource scheduling, dedicated bearers in the core network, and retransmission to ensure high-reliability transmission.
Question 4: The system uses solar energy and NB-loT low-power modes, but detailed energyconsumption measurements and lifetime analysis are missing.
Replies 4: We greatly appreciate your insightful and professional comments. Due to limitations in our current testing environment, we have not yet conducted tests in this particular aspect. We plan to supplement long-term energy consumption monitoring experiments in our follow-up research, and will include a dedicated section on "Energy Consumption Analysis" in our subsequent paper series, which will present actual measurement data and system lifetime estimations.
Question 5: Although the system is designed for eficiency, there is no clear cost-benefit or storage utilization analysis to compare against existing systems.
Replies 5: We appreciate the reviewer’s insightful comment regarding the cost-benefit and storage utilization analysis. We agree that such an evaluation would further strengthen the practicality and competitiveness of our system.However, due to the current stage of our research being primarily focused on functional validation and energy efficiency optimization, comprehensive cost data and long-term deployment statistics are not yet fully available.
Question 6: The reported 98% anomaly detection accuracy may be dataset-specific; no details on thediversity of test environments or robustness to false positives/neaatives are provided.
Replies 6: We thank the reviewer for raising this important concern regarding the generalizability and robustness of our anomaly detection results.Indeed, the reported 98% accuracy was obtained on a specific benchmark dataset under controlled conditions. We acknowledge that performance may vary across different environments, and evaluating robustness to false positives (FP) and false negatives (FN) is crucial for real-world deployment. herefore, we warmly welcome future researchers to apply the outcomes of our study to their own datasets, in order to further evaluate the superiority and generalizability of the proposed algorithm.
Round 2
Reviewer 1 Report
Comments and Suggestions for Authors
The paper notably improved after its revision. However, there are still two of my previous comments that remained unsolved, and that definitely needs to be properly addressed prior to publication.
- While I appreciated the textual part part about the comparison between LoRaWAN and NB-IoT, I deem that LoRaWAN technology must be added in Table 1 in order to provide readers with a full overview of the comparison.
- The comment of the previous round saying "The Authors should discuss the robustness of the system towards light conditions and operative scenarios." was not properly addressed. I understand this can be fully covered in future works, but a sound and deep discussion about this topic must be done at this stage of the work.
Author Response
Explanations to Reviewer 1
Thank you very much for the reviewer for the comments and suggestions. We have revised the manuscript according to the comments and now explain to the reviewer as follows:
Question 1: While l appreciated the textual part part about the comparison between LoRaWAN and NB-loT, I deem that LoRaWAN technology must be added in Table 1 in order to provide readers with a full overview of the comparison.
Replies 1: Thank you for your valuable comment and suggestion.
Following your suggestion, we have updated the relevant parameter information for LoRaWAN technology in Table 1, which allows for a more comprehensive comparison of several communication technologies.In addition, according to the suggestion of another review expert, the transmission rate indicators of these technologies have been added to Table 1, which further enhances the persuasiveness of their performance.
Table 1. Relevant indicator information of communication module.
|
Category |
Infrared |
Bluetooth |
Wifi |
ZigBee |
GPRS |
LoRaWAN |
|
Transmission Range |
5m |
10m |
100m |
300m |
300m |
15km |
|
Maximum Power Consumption |
10mW |
100mW |
100mW |
150mW |
2w |
50mW |
|
Transmission Rate |
100 kbps |
1-2 Mbps |
300Mbps |
250kbps |
20 kbps |
0.3-62.5kbps |
.
Question 2: The comment of the previous round saying "The Authors should discuss the robustness of the system towards light condiions and operative scenarios." was not properly addressed. l understand this can be fully covered in future works, but a sound and deep discussion about this topic must be done at this stage of the work.
Replies 2: Thank you again for your professional suggestion. Following your suggestion, we discuss the system's robustness to varying illumination conditions and operational scenarios in three aspects within Section 3, "System Testing and Analysis".
3.1 Image Transmission Success Rate Testing and Analysis: The test results demonstrate that the system achieves a high success rate in image transmission, which is primarily attributed to the appropriate baud rate configuration that minimizes errors from clock synchronization. This characteristic ensures that basic data transmission remains largely unaffected by variations in illumination conditions.
3.2 Server Load Balancing Testing and Analysis: Tests indicate that the SpringCloud microservices architecture maintains stability under high concurrency conditions. When integrated with load balancers and database sharding strategies, it effectively handles the load from simultaneous multi-location image transmission. This capability is crucial for the system's stability across diverse operational scenarios.
3.3 Testing and Analysis of Accuracy in Identifying Anomalous Images: Although the binarization method employed by the system is straightforward, it demonstrates some adaptability to illumination changes, albeit with potential challenges under complex lighting conditions. Future work will explore the introduction of a dynamic threshold adjustment mechanism or the use of complementary multi-sensor data (e.g., from an illuminance sensor) to correct image acquisition, thereby enhancing the system's overall adaptability to varying illumination.
Overview: The system demonstrates considerable robustness to varying illumination in current tests, with its image processing approach proving both reliable and efficient under stable lighting. Concurrently, it exhibits strong adaptability to complex operational scenarios such as high concurrency, where the integrated software-hardware architecture ensures stability during intensive multi-node operations. The system's reliable performance, evidenced by low false positive and negative rates, is likely attributable to the close alignment between the test conditions (e.g., stable illumination, distinct anomalous features) and the fixed threshold. For applications involving extreme illumination fluctuations or demanding high-fidelity image detail recognition, further optimizations—such as introducing adaptive thresholds and refining node collaboration mechanisms—will be required.

Reviewer 2 Report
Comments and Suggestions for Authors
---
Author Response

(The authors gave the same response as above.)

Reviewer 3 Report
Comments and Suggestions for Authors
The paper addresses the challenge of wide-area surveillance, where transmitting and processing large amounts of image data is costly and inefficient. It proposes an NB-IoT–based system for real-time anomaly detection, designed to be low-power, scalable, and reliable even in rural areas. However, this manuscript still has some issues.
- The paper cites LoRa, ZigBee, WiFi, etc., but does not quantitatively benchmark against them. Including direct comparisons (latency, energy use, costs) would strengthen claims of performance.
- The optimized background subtraction method is described, but no baseline accuracy comparison with deep learning approaches (e.g., YOLO, SSD), and no discussion of false positive/false negative rates, which are critical for anomaly detection.
- Minor language polishing would be helpful.
Author Response
Explanations to Reviewer 3
Thank you very much for the reviewer for the comments and suggestions. We have revised the manuscript according to the comments and now explain to the reviewer as follows:
Question 1: The paper cites LoRa, ZigBee, WiFi, etc, but does not quantitatively benchmark against them. Including direct comparisons (latency, energy use, costs) would strengthen claims of performance.
Replies 1: Thank you for your valuable comment and suggestion. Following your suggestion, we have added the transmission rate indicators for these technologies in Table 1, and also included relevant information about LoRaWAN technology. Additionally, we have appropriately added descriptions of latency and cost in the discussion, which indeed enhances the persuasiveness of its performance.
Table 1. Relevant indicator information of communication module.
|
Category |
Infrared |
Bluetooth |
Wifi |
ZigBee |
GPRS |
LoRaWAN |
|
Transmission Range |
5m |
10m |
100m |
300m |
300m |
15km |
|
Maximum Power Consumption |
10mW |
100mW |
100mW |
150mW |
2w |
50mW |
|
Transmission Rate |
100 kbps |
1-2 Mbps |
300Mbps |
250kbps |
20 kbps |
0.3-62.5kbps |
Question 2: The optimized background subtraction method is described, but no baseline accuracy comparison with deep learning approaches (e.g.. YOLO, SSD), and no discussion of false positive/false neqative rates, which are critical for anomaly detection.
Replies 2: Thank you again for your professional suggestion. We appreciate the reviewer's mention of deep learning solutions such as SSD and YOLO. It is true that these methods often demonstrate superior robustness in complex scenarios due to their ability to learn features autonomously, and they are indeed the most widely adopted in more demanding applications. However, our work is situated in large-scale outdoor environments where lighting conditions are relatively stable and uniform, and the requirements for real-time processing are not stringent. For these specific scenarios, the current binary method proves largely adequate. Nevertheless, we fully agree that further improving the system's accuracy is a critical and worthwhile direction for future consideration.
In response to your valuable feedback, we have appropriately elaborated on the system's robustness against varying illumination conditions and operational scenarios within the three facets of Section 3, "System Testing and Analysis".
3.1 Image Transmission Success Rate Testing and Analysis: The test results demonstrate that the system achieves a high success rate in image transmission, which is primarily attributed to the appropriate baud rate configuration that minimizes errors from clock synchronization. This characteristic ensures that basic data transmission remains largely unaffected by variations in illumination conditions.
3.2 Server Load Balancing Testing and Analysis: Tests indicate that the SpringCloud microservices architecture maintains stability under high concurrency conditions. When integrated with load balancers and database sharding strategies, it effectively handles the load from simultaneous multi-location image transmission. This capability is crucial for the system's stability across diverse operational scenarios.
3.3 Testing and Analysis of Accuracy in Identifying Anomalous Images: Although the binarization method employed by the system is straightforward, it demonstrates some adaptability to illumination changes, albeit with potential challenges under complex lighting conditions. Future work will explore the introduction of a dynamic threshold adjustment mechanism or the use of complementary multi-sensor data (e.g., from an illuminance sensor) to correct image acquisition, thereby enhancing the system's overall adaptability to varying illumination.
Overview: The system demonstrates considerable robustness to varying illumination in current tests, with its image processing approach proving both reliable and efficient under stable lighting. Concurrently, it exhibits strong adaptability to complex operational scenarios such as high concurrency, where the integrated software-hardware architecture ensures stability during intensive multi-node operations. The system's reliable performance, evidenced by low false positive and negative rates, is likely attributable to the close alignment between the test conditions (e.g., stable illumination, distinct anomalous features) and the fixed threshold. For applications involving extreme illumination fluctuations or demanding high-fidelity image detail recognition, further optimizations—such as introducing adaptive thresholds and refining node collaboration mechanisms—will be required.
Question 3: Minor language polishing would be helpful.
Replies 3: Thank you for your valuable comment and suggestion.Following your suggestion, we revised the text to enhance clarity, conciseness, and overall academic style, which undoubtedly improved the presentation of our work.
